# Biochemical Analyses of Bioactive Extracts from Plants Native to Lampedusa, Sicily Minor Island

**DOI:** 10.3390/plants11243447

**Published:** 2022-12-09

**Authors:** Roberta Di Lecce, Natacha Mérindol, Mayra Galarza Pérez, Vahid Karimzadegan, Lionel Berthoux, Angela Boari, Christian Zidorn, Maurizio Vurro, Giuseppe Surico, Isabel Desgagné-Penix, Antonio Evidente

**Affiliations:** 1Department of Chemical Sciences, University of Naples Federico II, Complesso Universitario Monte Sant’Angelo, 80126 Napoli, Italy; 2Département de Chimie, Biochimie et Physique, Université du Québec à Trois-Rivières, Trois-Rivières, QC G8Z 4M3, Canada; 3Pharmazeutisches Institut, Abteilung Pharmazeutische Biologie, Christian-Albrechts-Universität zu Kiel, Gutenbergstraße 76, 24118 Kiel, Germany; 4Département de Biologie Médicale, Université du Québec à Trois-Rivières, Trois-Rivières, QC G8Z 4M3, Canada; 5Institute of Sciences of Food Production, National Research Council, Via Amendola 122/O, 70125 Bari, Italy; 6Department of Agriculture, Food, Environment, and Forestry (DAGRI), Section of Agricultural Microbiology, Plant Pathology and Enthomology, University of Florence, 50121 Firenze, Italy

**Keywords:** Lampedusa island, Sicily, native plants, metabolic profile, biological activity screening, antiviral, biopesticide

## Abstract

Major threats to the human lifespan include cancer, infectious diseases, diabetes, mental degenerative conditions and also reduced agricultural productivity due to climate changes, together with new and more devastating plant diseases. From all of this, the need arises to find new biopesticides and new medicines. Plants and microorganisms are the most important sources for isolating new metabolites. Lampedusa Island host a rich contingent of endemic species and subspecies. Seven plant species spontaneously growing in Lampedusa, i.e., *Atriplex halimus* L. (*Ap*), *Daucus lopadusanus* Tineo (*Dl*), *Echinops spinosus* Fiori (*Es*) *Glaucium flavum* Crantz (*Gf*) *Hypericum aegypticum* L: (*Ha*), *Periploca angustifolia* Labill (*Pa*), and *Prasium majus* L. (*Pm*) were collected, assessed for their metabolite content, and evaluated for potential applications in agriculture and medicine. The HPLC-MS analysis of *n*-hexane (HE) and CH_2_Cl_2_ (MC) extracts and the residual aqueous phases (WR) showed the presence of several metabolites in both organic extracts. Crude HE and MC extracts from *Dl* and *He* significantly inhibited butyrylcholinesterase, as did WR from the extraction of *Dl* and *Pa*. HE and MC extracts showed a significant toxicity towards hepatocarcinoma Huh7, while *Dl*, *Ha* and *Er* HE extracts were the most potently cytotoxic to ileocecal colorectal adenocarcinoma HCT-8 cell lines. Most extracts showed antiviral activity. At the lowest concentration tested (1.56 μg/mL), *Dl, Gf* and *Ap* MC extracts inhibited betacoronavirus HCoV-OC43 infection by> 2 fold, while the *n*-hexane extract of *Pm* was the most potent. In addition, at 1.56 μg/mL, potent inhibition (>10 fold) of dengue virus was detected for *Dl, Er,* and *Pm* HE extracts, while *Pa* and *Ap* MC extracts dampened infections to undetectable levels. Regarding to phytotoxicity, MC extracts from *Er*, *Ap* and *Pm* were more effective in inhibiting tomato rootlet elongation; the same first two extracts also inhibited seed cress germination while its radicle elongation, due to high sensitivity, was affected by all the extracts. *Es* and *Gf* MC extracts also inhibited seed germination of *Phelipanche ramosa*. Thus, we have uncovered that many of these Lampedusa plants displayed promising biopesticide, antiviral, and biological properties.

## 1. Introduction

Nature has always been an important source of novel compounds with original carbon skeletons whose biological activities show potential applications in different fields, particularly in agriculture and medicine. The major sources of bioactive metabolites have been terrestrial or marine living organisms, including plant pathogenic or endophytic microbes and marine microorganisms [1]. Plants are a rich reservoir of secondary metabolites that can be used as defense, hormones or to attract pollinators, and can be used in food chemistry, agriculture, and medicine. The plant secondary metabolites could be grouped according to their structure and/or biosynthetic origin or chemical-physical common properties. Alkaloids are one of the biggest groups of natural compounds containing basic nitrogen, that can be in turn grouped into different families on the basis of their chemical structure. Other groups of plant metabolites are those that originated from the shikimic acid biosynthetic pathway as the two aromatic amino acids phenylalanine and tyrosine, and their by-products such as cinnamic acids, coumarins, lignans and lignins, phenylpropanoids and benzenoids derivatives. In addition, there are the groups derived from the acetate biosynthetic pathway as polyketides, including naphthoquinones and anthraquinones, their analogs, and oligomers. The largest class of naturally occurring compounds is constituted of the terpenes, which could be sub-grouped according to the number of isoprene units, which are linked according to rigid biosynthetic rules, and are included in their carbon skeleton as monoterpenes, sesquiterpenes, diterpenes, sesterpenes, triterpenes, tetraterpenes, and polyisoprene [2,3,4]. Among all these classes of natural compounds, there are metabolites that are anti-inflammatory, antioxidant, anticancer, antimicrobial, antiviral, and antidiabetic, etc., with potential application in medicine [2]; as well as biopesticides, including fungicides, herbicides, insecticides, nematocides etc., with potential application in agricultures [5]; and neutraceuticals, colors, aromas, etc., with potential application in food chemistry [6]. Among these there are alkaloids isolated from Amaryllidaceae and alkylamides [7,8], with the latter being a broad and expanding group found in at least 33 plant families [8]. Alkylamides and alkaloids have been observed to have antimicrobial [9,10,11,12,13,14], antiprotozoal [11,15], antiviral [16,17], insecticidal [18,19,20,21], nematocidal [22], cytotoxic [17], anticancer [23,24,25,26,27,28,29,30,31,32], acetylcholinesterase inhibition [27,28], and bioherbicidal activities [20,30]. Some plant extracts with potential herbicidal activity were also appropriately formulated in biopolymers [31,32]. Plant extracts can display significant activity on their own, or be sources of antimicrobial compounds effective against human infectious [33,34,35].

Here, we performed a screening of endemic plants present in different regions of the Mediterranean basin in order to isolate new bioactive extracts from species collected in Lampedusa, which together with Linosa and Lampione, form the Pelagie islands Archipelago in the Mediterranean Sea.

Seven plant species spontaneously growing in Lampedusa, specifically *Atriplex halimus* L., *Daucus lopadusanus* Tineo, *Echinops spinosus* Fiori, *Glaucium flavum* Crantz, *Hypericum aegypticum* L., *Periploca angustifolia* Labill and *Prasium majus* L., were collected, extracted, summarily investigated for their metabolite content, and evaluated for their potential applications in agriculture and medicine.

## 2. Results and Discussion

The crude extracts of the seven fresh plants collected in Lampedusa island, namely *Atriplex halimus* L. *(Ap), Daucus lopadusanus* Tineo *(Dl), Echinops spinosus* Fiori *(Es), Glaucium flavum* Crantz *(Gf), Hypericum aegypticum* L. *(Ha), Periploca angustifolia* Labill *(Pa),* and *Prasium majus* L. *(Pm)* (see Appendix A) were obtained as detailed in the Materials and Methods. Plants were first extracted with *n*-hexane (HE) and then with methylene chloride (MC) and the resulting organic extracts were used for chemical and biological investigations. Table 1 summarizes the plant source and botanic family, the yields of the two organic residues obtained by plant successive extraction with HE and DC, and the relevant literature found on SciFinder for each plant.

Based on literature data (Table 1), *Dl*, *Es* and *Gf* were the most promising plants for the isolation of new metabolites because no metabolites were previously reported for these species. *Ha* also seemed interesting, as only nine published studies were reported but any bioactive metabolite was characterized. One article described the content of the main polyphenols, acylphloroglucinols, and naphthodianthrones extracted and analyzed by high performance liquid chromatography with diode-array detection and mass spectrometry (HPLC-DAD-MS) [36]. One of the articles regarding *Pm* described the presence of phenols, flavonoids, and tannins [37], another the iridoids and flavonoid glycosides [38], and the third investigated terpenes in the essential oil [39].

Among the four references obtained for *Ap,* one reported the isolation of glycosyides of flavanoids and phenols [40] and another referred to a characterized metabolite, namely septanoside. In fact, a new septanoside and its new derivatives, named portulasoid and septanoecdysone, respectively, were isolated together with the known 20-hydroxyecdysone from Tunisian *Ap*. Some metabolites showed antioxidant, antibacterial, and anticholinesterase activities [41]. Septanosides, also named septanoses, are naturally occurring sugars having a seven-membered oxepane ring. Their chemical properties were recently reviewed [42]. Regarding *Pa,* four articles reported on the polyphenols, flavonoids, carotenoids, tannins and anthocyanins extracted from the different part of the plants [43,44,45]. Additionally, polysaccharides were biotechnological produced from this plant. However, any metabolite was isolated and characterized from this plant [46]. Thus, in order to increase our knowledge on these plants as well as chemically and biologically characterize their most interesting extracts, a preliminary chemical study and a deep biological investigation were performed.

The *n*-hexane (HE) and methylene chloride (MC) extracts and the residual aqueous phase (WR) were first analyzed for their metabolic profile by HPLC-MS. The chromatographic profiles for *Da* (1a–1d, 2a–2d, 3a–3d), for *Gf* (4a–4d, 5a–5d; 6a–6d), for *Ha* (7a–7d; 8a–8d; 9a–9d), for *Pa* (10a–10d; 11a–11d; 12a–12d) for *Es* (13a–13d; 14a–14d; 15a–15d) for *Pm* (16a–16d; 17a–17b; 18a–18b); *Ap* (19a–19d; 20a–20d; 21a–21d) are reported in SM (Appendix A). In particular, for each extract the base peak chromatograms (BPC) were recorded in both positive and negative ionization modes of WR, and the photodiode array detector PDA chromatogram of HE and MC extracts was obtained at both 245 and 360 nm.

As expected, the HPLC-MS chromatograms showed very polar compounds in the WR, low polar ones in HE and medium polarity compounds in the MC extracts. The profile obtained in positive ion mode showed a content of metabolite different from those recorded in the negative ion mode as well as those recorded at 254 and 360 nm for both the organic extracts and the WR. Thus, the results showed the presence of several metabolites in both organic extracts and there are noteworthy differences in the nature of the metabolites produced by the different plants.

As an example are reported the two PDA HPLC-MS chromatograms recorded at 245 nm of the HE and MC extracts of *D. lopadusanus* (Figure 1A,B). They differ for the presence in the first one (HE) of main metabolites with medium-low polarity and molecular weight ranging 100–600 mAU, while that of MC extract contain as main components more polar metabolites with molecular weight < 200 mAU and few metabolite with highest mAU.

The two organic extracts and the WRs were used to assay different biological activities. First, the anti-Alzheimer’s disease potential of the plant extracts and corresponding aqueous phase was evaluated by measuring the inhibition of butyrylcholinesterase and acetylcholinesterase catalyzed reactions. The well-known inhibitors rivastigmine and galanthamine were used as positive controls [47]. At a concentration of 100 μg/mL, all *Ha* extracts, *Dl* HE and DC extracts, and *Pa* water extract led to an inhibition of butyrylcholinesterase activity (>50%) (Figure 2A). This suggests that these plant extracts contain anti-butyrylcholinesterase compounds that should be individually investigated. Acetylcholinesterase inhibition was weak in all extracts, but at 100 μg/mL, MC extracts from *Dl*, *Gf*, *Ha*, *Pa*, *Es* and *Pm* led to a ~50% inhibition (Figure 2B).

Next, the cytotoxic potential of plant extracts was evaluated using the hepatocarcinoma Huh7 [48] and the ileocecal colorectal adenocarcinoma HCT-8 [49] cell lines. Overall, the HE and MC extracts were cytotoxic to the cell lines, suggesting an enrichment in cytotoxic metabolites. At 100 μg/mL, *Dl* and *Ha* HE extracts decreased HCT-8 cells viability by >50% of, while *Dl*, *Ha*, *Gf*, *Pa*, *Es* and *Ap* HE extracts decreased Huh7 cells viability by >50% (Figure 3A,B). 25 μg/mL of the HE fraction of *Dl*, *Gf*, *Pa*, *Es*, *Pm* and *Ap* also decreased Huh7 cells viability by ≥50% of, but had no effect on HCT-8, except for *Pm*. At 6.25 μg/mL, *Ap* HE fraction still decreased Huh7 viability by >50%, suggesting the presence of highly cytotoxic, cell line specific compounds. Of note, ethanol extracts of *Gf* were previously shown to be cytotoxic against HepG2 and HCT cells [50], but to our knowledge, the cytotoxicity of the other plant extracts fractions is reported here for the first time.

Emerging and reemerging infectious diseases threatening human health commonly originate from the betacoronavirus and the flavivirus genus [51,52]. We used the Amaryllidaceae alkaloid lycorine as a positive control, as it was shown to display a broad antiviral activity in previous studies [53,54]. We observed that many of the plant species included in this study possessed antiviral properties, especially for the organic extracts using non-polar and slightly polar HE and MC solvents. At the lowest concentration, 1.56 μg/mL, HCoV-OC43 infection was inhibited >2 fold by *Dl*, *Gf* and *Ap* MC extracts (Figure 3A). At 1.56 μg/mL, the HE extract of *Pa* was the most potent, leading to a 4-fold decrease in replication, while coronaviral inhibition was also detected in *Gf*, *Ha* and *Es* HE extracts. At 6.25 μg/mL, MC extracts of *Dl* and *Ap* reduced infection by >10-, and >100- fold, respectively. At that concentration, betacoronavirus replication was also significantly inhibited by the MC extracts of *Ha* and *Es*, while at ≥25 μg/mL DMC extracts of all plant species potently blocked infection. At 6.25 μg/mL HCoV-OC43 inhibition activity was also detected in HE extracts of *Pm*, and *Ap*, and at 25 μg/mL in *Dl* (2-fold inhibition). At 100 μg/mL all plant-derived HE and MC extracts lead to a ~100-fold (background levels of infection) inhibition of coronaviral replication. Interestingly, the aqueous phase of *Ha*, *Pa* and *Ap* also lead to a detectable-to-strong inhibition at that concentration (Figure 4A).

Similarly, in the case of DENV replication, treatment with 100 μg/mL of HE or MC extracts from all plant species lead to a potent inhibition (Figure 4B). A ~100-fold inhibition was observed using MC extracts from *Dl*, *Ha*, *Pa*, *Er* and *Ap*, and a >10-fold inhibition was seen with extracts from *Gf* and *Pm*. Antiviral activity was also detectable in HE extracts from *Gf*, *Ha*, *Pa*, *Ers*, *Pm* and *Ap* at 100 μg/mL, albeit at lower levels compared to MC in most cases. At 25 μg/mL, a potent inhibition (>10 fold) was detected with *Dl, Ha,* and *Pm* HE extracts, while *Pa*, and *Ap* CM fractions blocked infection to background levels (~100 fold). Remarkably, *Pa* WR phase also inhibited DENV infection at 25 and 100 μg/mL. The results suggest the presence of broad-spectrum anti-RNA virus molecules in many of the tested plant extracts.

At non-cytotoxic concentrations, the extracts that most potently inhibited coronavirus replication were the HE extracts from *Pa*, and the MC fraction from *Ap* and *Dl.* For DENV, HE extracts of *Ha,* and *Pa* WR showed the strongest antiviral activities. Interestingly, specialized metabolites (specifically napthodianthrones) previously isolated from other *Hypericum* species were shown to display some anti-HIV-1 potential (hypericin), or anti-coronaviral activity (SARS-CoV-2, quercetin and hypericin), while others (acylphloroglucinol derivatives) were shown to be cytotoxic [55,56,57,58].

We then assessed the anti-germination ability of the extracts on tomato seedlings. MC extracts proved to be, on average, more effective than HE extracts and WR (Table 2). In particular, extracts from *Es* (Figure 5) *Pm* and *Ap* were more effective at inhibiting rootlet elongation. On cress, seed germination was inhibited particularly by the MC extracts from *Es* and *Ap*, whereas all the other extracts were weakly or not active. In the assay on cress radicle elongation, almost all the extracts were effective regardless of origin, confirming that this plant is very sensitive to herbicidal compounds; indeed, it is frequently used as reference plant to detect residues of herbicides present in soil. In the assay on broomrape seeds, MC extracts were also the most active. In particular, the extracts from *Es* (Figure 6) and *Gf* were able to reduce germination of the stimulated seeds by approximately 50% compared with the control.

Applied on the leaf surface by puncture (Table 3), as expected the extracts caused only modest (if any) effects, consisting of small necrotic or chlorotic spots, limited to the area of application of the droplets. On average, the MC extracts were more active than HE or WR residues extracts, with the exception of *Pa* on *He* MC extracts, which proved to be completely inactive.

Finally, assayed on three fungal species, only the MC extracts obtained from *Ap* showed a modicum antifungal activity against *Penicillium italicum*, but not against the other two fungal species (data not shown). All the other extracts, both from MC as well as from HE, were completely inactive.

## 3. Materials and Methods

### 3.1. General Procedures

Analytical TLCs were performed on silica gel plates (Merck, Kieselgel 60F254, 0.25 and 0.5 mm, Merck, Darmstadt, Germany). The spots were visualized by exposure to UV light and/or iodine vapours and/or by spraying with 10% H_2_SO_4_ in MeOH and with 5% phosphomolybdic acid in EtOH, followed by heating at 110 °C for 10 min. Sigma Aldrich Co. (St. Louis, MO, USA, supplied all reagents and solvents. LC-MS grade acetonitrile, water, and other (analytical grade) solvents were obtained from VWR International GmbH, Darmstadt, Germany. Dimethyl sulfoxide was purchased from, Carl Roth GmbH, Karlsruhe, Germany.

### 3.2. Plant Material

Whole aerial parts of *A. halimus* L., *D. lopadusanus* Tineo, *E. spinosus* Fiori, *G. flavum Crantz*, *H. aegypticum* L., *P. angustifolia Labill*, and *P. majus* L. plant samples [59,60,61] were fresh collected on 14–19 April 2022 in different sites of Lampedusa island (Italy) by Fabio Giovanetti and identified by Giuseppe Surico, University of Florence, Italy. The plant specimens are deposited in the collection of Department of Agriculture, Food, Environment, and Forestry (DAGRI), Section of Agricultural Microbiology, Plant Pathology and Entomology, University of Florence, Italy, n. DAGRI-56.

### 3.3. General Procedure for the Extraction of Plant Metabolites

Plant materials (200 g) were extracted (1 × 0.5 L) by H_2_O/MeOH (1/1, *v*/*v*) under stirred conditions at room temperature for 24 h, and the obtained suspensions centrifuged at 7000 rpm at 5 °C. Supernatants were separated from the solid phase and extracted by *n*-hexane (3 × 200 mL). The organic extracts were combined, dried (Na_2_SO_4_), filtered and evaporated under reduced pressure, yielding an oily residue. The residual aqueous phase was extracted with CH_2_Cl_2_ (3 × 200 mL). The organic extracts were combined, dried (N_2_SO_4_), filtered and evaporated under reduced pressure, yielding an oily residue. The weight of the two oily residues obtained from each plant extraction are reported in Table 1.

### 3.4. HPLC-MS Analyses of Plant Organic Extracts

#### 3.4.1. Samples Preparation

*n*-Hexane and dichloromethane dry extracts were dissolved in acetone; aqueous phases were dissolved in dimethyl sulfoxide. Solutions of 2 mg per mL were prepared. 

#### 3.4.2. UHPLC-DAD-MS Analyses

UHPLC-DAD-MS analyses were carried out on a Shimadzu Nexera 2 liquid chromatograph with a LC30AD binary pump, connected to a SIL-30AC autosampler, CTO-20AC column heater, SPD-M30A diode Array detector and a LC-MS 8030 triple quadrupole mass spectrometer using electrospray ionization (Shimadzu, Kyoto, Japan). The column used was a Phenomenex Luna Omega C18 (100 × 2.1 mm, 1.6 µm particle size) (Phenomenex, Aschaffenburg, Germany).

For analyses, 0.1% formic acid in LC-MS water (A) and acetonitrile (B) were used as solvents with the following gradient: 5% B to 95% B in 45 min held for 15 min. The post-run time was set to 10 min, the temperature to 30 °C. The flow rate was 0.2 mL/min. The injection volume was 2.00 μL. Base Peak Chromatograms (BPC) were recorded in the positive and negative ionization mode for a mass range of *m*/*z* 200–1500 and *m*/*z* 200–1000 respectively. PDA chromatograms were recorded at 360 and 254 nm.

#### 3.4.3. Chemical Reagents

LC-MS grade formic acid was purchased from Sigma Aldrich Co., St. Louis, MO, USA. LC-MS grade acetonitrile, water, and other (analytical grade) solvents were obtained from VWR International GmbH, Darmstadt, Germany. Dimethyl sulfoxide was purchased from, Carl Roth GmbH, Karlsruhe, Germany.

### 3.5. Biological Assays

#### 3.5.1. Cytotoxic Assays

CH_2_Cl_2_ and *n-*hexane plant extracts were resuspended in DMSO, and diluted with H_2_O at a final concentration of 10 mg/mL. Huh7 cells (ATCC) were maintained in Dulbecco’s Modified Eagle Medium (DMEM) supplemented with 10% fetal bovine serum (FBS) and 1% penicillin/streptomycin solution (all from Wisent, Inc., Saint Bruno, QC, Canada). HCT-8 cells (ATCC) were maintained in Roswell Park Memorial Institute (RPMI) medium supplemented with 10% horse serum (HS) and 1% penicillin-streptomycin solution (all from Thermo Fisher, Inc., Mississauga, ON, Canada). Cell lines were kept in an incubator at 37 °C and 5% CO_2_. Cytotoxicity assays of plant extracts were performed by using the MTT-based colorimetric assay (Cell proliferation Kit I, Roche, Millipore Sigma, Merck, Darmstadt, Germany). Briefly, 7.5 × 10^3^ Huh7 cells/well or 2 × 10^4^ HCT-8 cells/well were plated in 96-well plates and incubated at 37 °C overnight. The next day, plant extracts, DMSO, H_2_O (as solvent controls) and lycorine (as cytotoxic control, see lit. n., [62]) were serially diluted by a factor of 4 in DMEM (for Huh7) or RPMI complete medium (for HCT-8) at room temperature. Each dilution was added to the cells at final concentrations of 1.56, 6.25, 25, and 100 µg/mL for plant extracts, and 0.78 to 25 µg/mL for lycorine. The plates were then incubated at 37 °C and 5% CO_2_ for 24 (Huh7) or 72 h (HCT-8). The MTT labeling reagent (final concentration 0.5 mg/mL) was added, followed by the solubilizing solution 4 h later. Plates were then incubated overnight, according to the manufacturer instructions. Spectrophotometrical absorbance of the formazan product was read at 575 nm on a microplate spectrophotometer (Synergy H1, Biotek, QC, Canada), using 700 nm as reference wavelength. Medium only (no cells) with MTT reagent was used to subtract background and the % of viable cells was calculated as the ratio of absorbance in cells in presence of extract over cells in absence of extract (maximal viability).

#### 3.5.2. Antiviral Assays

##### Human Coronavirus-OC43 (HCOV-OC43) Infection

Briefly, 2 × 10^4^ HCT-8 cells (HRT-18, CCL24 from ATCC), were seeded in 96-well plates, treated with plant etracts and controls and infected with HCoV-OC43 (Betacoronavirus 1, VR1558, ATCC) at an MOI = 0.01, as representative of the Betacoronavirus genus. Cells were incubated for 5 days, fixed with 4% formaldehyde and permeabilized with 0.1% Triton X-100. A monoclonal mouse antibody specific for the HCoV-OC43 nucleoprotein (clone 542-7D; Sigma-Aldrich) and chicken anti-mouse immunoglobulin conjugated to CF^TM^ 488 (Millipore Sigma) were used as the primary and secondary antibodies, respectively. Wells were washed three times between each of the fixation, permeabilization and staining with antibody steps with PBS buffer. Incubation in the primary antibody (2 μg/mL) was done overnight at 4 °C while incubation in the secondary antibody (5 μg/mL) was for 60 min at room temperature. All antibodies were diluted in PBS buffer containing 0.5% bovine serum albumin. After washing three times with PBS, the % of infected fluorescent cells was analyzed on a Beckman Cytoflex flow cytometer. At least 10,000 events were acquired. Data analysis was performed using the Flowjo software (BD, FlowJo LLC, Ashland, OR, USA). The fold inhibition of viral infection was calculated as the ratio of the % of untreated infected cells (maximal infection) over the % of infected cells in presence of extract.

##### Dengue Virus Infection

We used green fluorescent protein (GFP)-encoding dengue virus propagative vector (DENV_GFP_) as representative of the flavivirus genus. The plasmid used to obtain the DENV_GFP_ vector (pFK-DVs-G2A) was provided by Ralf Bartenschlager (Heidelberg University, Heidelberg, Germany) and Laurent Chatel-Chaix (Institut national de la recherche scientifique, Québec, QC, Canada) [63,64]. For DENV_GFP_, viral titer was measured by plaque assay in VeroE6 cells, as described by [65]. Briefly, Huh7 cells/well (kindly provided by Pr Hugo Soudeyns) were plated in 96-well plates at 37 °C overnight, treated with plant extracts and controls, i.e., DMSO, H_2_O and lycorine (a known anti-DENV alkaloid [66,67] and infected with at a multiplicity of infection (MOI) of 0.1. Cells were incubated at 37 °C and 5% CO_2_ for 48 h. Afterwards, the cells were fixed in 3.7% formaldehyde and the percentage of infection was measured by flow cytometry with a FC500 MPL cytometer (Beckman Coulter, Inc., Brea, CA, USA). Data analysis was performed using the Flowjo software (BD, FlowJo LLC, Ashland, OR, USA).

#### 3.5.3. Anti-Cholinesterase Assays

Pharmacological properties specific to AD were tested by measuring butyrylcholinesterase (BuChE, equine, Millipore Sigma, Merck, Darmstadt, Germany) and acetylcholinesterase (AChE, electric eel, Millipore Sigma, Merck, Darmstadt, Germany) activity inhibition in a colorimetric assay using DTNB (Bis(3-carboxy-4-nitrophenyl) disulfide, Ellman’s Reagent, Millipore Sigma). The reaction was performed in a final volume of 100 μL phosphate buffer (0.1 M, pH = 7.5) in 96-well microplates. H_2_O and DMSO-dissolved extracts were added to a final concentration of 100, 10 or 1 μg/mL (1, 0.1 and 0.01% DMSO or water). Next, a 5 μL reaction mixture containing equal amounts of acetylthiocholine (20×) (Millipore Sigma) and DTNB (20 × 0.01 M for BuChE and 0.0025 M for AChE assays) was added to each well. Finally, the enzyme solution was added to a final concentration of 0.125 U/mL for AChE and 2 U/mL for BuChE and absorbance was measured at 412 nm in kinetic mode for 10 min using a microplate reader (Synergy H1, Biotek, QC, Canada). Galanthamine (5.75, 0.57, 0.057 μg/mL) and rivastigamine (100, 10, 1 μg/mL) were used as a positive control for AChE and BuChE assays, respectively. Experiments were performed twice. BuChE inhibition was calculated as follows [68]:I = 100 × (1 − Δi/Δe)(1)
where Δi is the difference of absorbance between two time points in presence of the putative inhibitor, and Δe is the difference of absorbance using two time points in presence of DMSO or H_2_O. AChE inhibition was calculated using a one time-point ratio: 

I = 100 × (E−S)/E where E corresponds to the absorbance in absence of inhibitor and S represents the absorbance related to the test inhibitors. Data were normalized using DMSO reads.

#### 3.5.4. Statistical Analyses

All the analyses and related graphs werecarried out using the GraphPad Prism version 8.0.0 (GraphPad Software, San Diego, CA, USA).

#### 3.5.5. Phytotoxicity Bioassays

The *n*-hexane and methylene chloride extracts were suspended in MeOH at a concentration of 20 µg/µL. The aqueous phases were dissolved in water at a concentration of 1 µg/µL.

#### 3.5.6. Bioassays on Tomato Seeds

Seeds were sterilized for 10 min using a 1% aqueous solution of sodium hypochlorite, then rinsed in sterile water and left in large Petri dishes for two days to germinate. Then, germinated seeds (10 seeds with a radicle length of approx. 0.5 cm) were placed in Petri dishes (diam. 60 mm) on a double paper filter soaked with 1.5 mL of the solution to be tested (1% the final concentration of MeOH for the *n*-hexane and methylene chloride extracts). Three days after the application, the length of the radicle was measured and compared with that of the control (water). Two replicates were prepared for each solution

#### 3.5.7. Bioassays on *Phelipanche ramosa* Seeds

The bioassay was performed as described previously [69] except that 6-well plates were used. The filters were soaked with 1 mL of the test solution with 0.5% MeOH for the *n*-hexane and methylene chloride extracts. Six days after treatment, the number of germinated seeds was determined and compared with the control (water with stimulant). Two replicates were prepared for each solution.

#### 3.5.8. Bioassays on Cress Seeds

After being rinsed in sterile water, ten seeds were placed in each well in 6-well plates, on filters previously soaked with 1 mL of each solution (1% MeOH concentration for the organic extracts). After three days, the number of germinated seeds were counted and compared with the control (distilled water). Two replicas were performed for each solution.

#### 3.5.9. Leaf Puncture Assay

The test was carried out by applying one droplet (30 µL, 4% MeOH) of the solution to detached leaves punctured with a needle immediately before the application. Leaves were kept in moistened chambers for three days and visually observed for the eventual appearance of phytotoxic symptoms (chlorosis or necrosis). Leaves of following weed species were harvested and tested: *Chenopodium album* L., *Chrozophora tinctoria* L., *Heliotropium europaeum* L., *Solanum nigrum* L., *Sonchus oleraceus* L., *Sorghum halepense* L., *Portulaca oleracea* L.

#### 3.5.10. Antifungal Bioassays

Fungal conidia suspensions (100 µL) were uniformly distributed onto Petri dishes (9 cm) containing a thin layer of PDA (12 mL). Then, 20 µL of the extracts (at a concentration of 20 µg/µL) were loaded on antimicrobial susceptibility test discs (Oxoid Ltd., Basingstoke, UK). Plates were kept in an incubator (25 °C) for two days, and the eventual appearance of a growth inhibition area around the disks was observed. Only the organic extracts and not the water extracts were tested in this assay. The following microscopic fungi were used, available in the ISPA fungal collection (in brackets the amount of conidia and the collection number: *Penicillium italicum* (1 × 10^6^ c/mL-9571), *Aspergillus carbonarius* (1 × 10^6^ c/mL-7462), *Drechslera gigantea* (1 × 10^4^ c/mL-7004)

## 4. Conclusions

In conclusion, HPLC-MS analysis of *n*-hexane and DCM extracts and the residual aqueous phases showed the presence of several and structurally different metabolites. The biological assays showed that both *n*-hexane and/or MC extracts from all the plants contain bioactive metabolites as found by testing their cytotoxicity against human cell lines. Furthermore, *Ha*, *Dl* and *Pa* extracts significantly inhibited butyrylcholinesterase, while those from *Dl, Gf* and *Ap* inhibited HCoV-OC43 infection >2-fold, with *Pm* being the most potent. Extracts from *Dl, Er* and *Pm* inhibited DENV, with *Pa*, and *Ap* extracts being the most efficient. MC extracts from *Er*, *Ap,* and *Pm* were the most effective in inhibiting tomato rootlet elongation; the same first two extracts inhibited seed cress germination while the radicle elongation, due to his high sensitivity, was affected by all the extracts. In addition, MC extracts from *Er* and *Gf* inhibited the stimulated *Orobanche ramosa* seeds germination. These results suggested to further investigate the organic extract of *Dl*, *Gf*, *Ap,* and *Pm* among all the spontaneous plants collected in Lampedusa Island.

## Figures and Tables

**Figure 1 plants-11-03447-f001:**
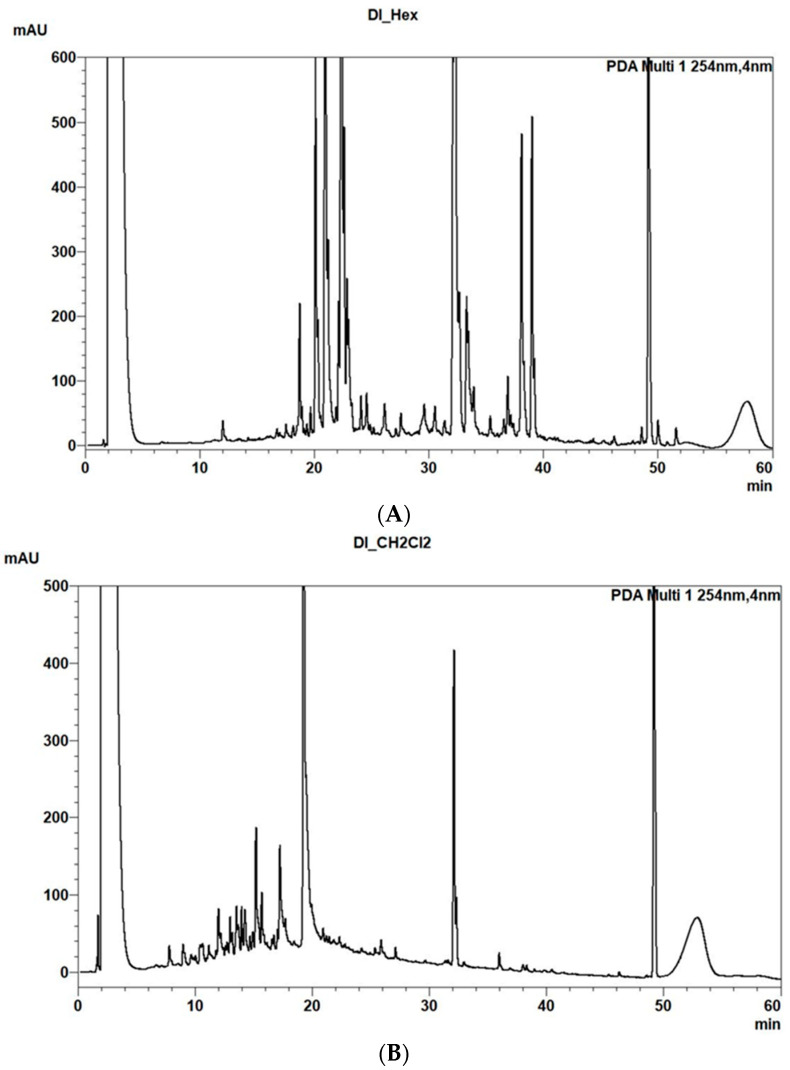
(**A**) PDA (Photodiode Array Extension) chromatogram of HE extract of *Daucus lopadusanus* recorded at 245 nm. (**B**) PDA chromatogram of MC extract of *Daucus lopadusanus* recorded at 245 nm.

**Figure 2 plants-11-03447-f002:**
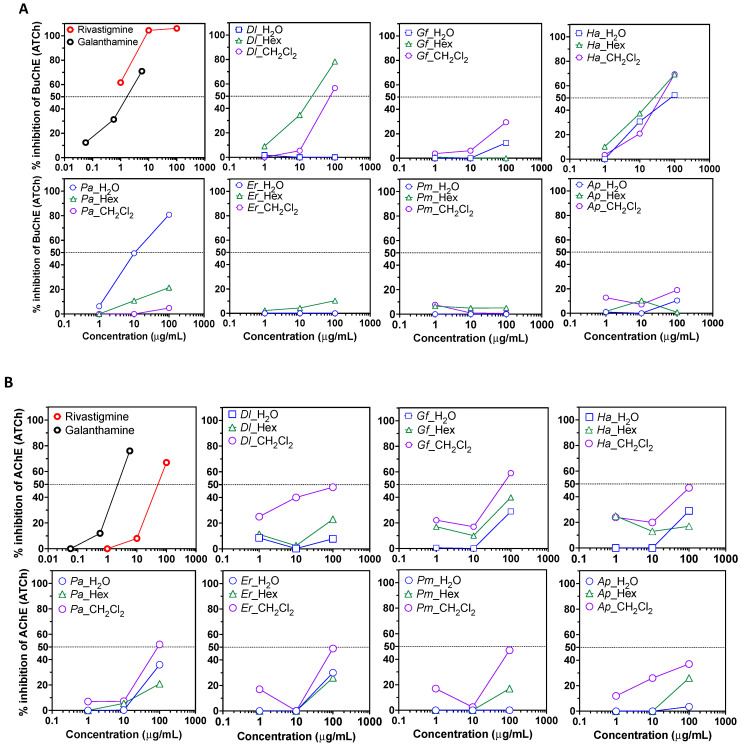
Anti-cholinesterase potential of plant extract fractions. (**A**) Butyrylcholinesterase (BuChE) inhibition assay using acetylthiocholine (ATCh) as a substrate. (**B**) Acetylcholinesterase inhibition assay. Data are presented as a ratio of absorbance over the corresponding negative control (DMSO or H_2_O solvants) at each concentration. Shown are representative results of at least two independent experiments. *Dl: Daucus lopadusanus*; *Gf: Glaucium flavum*; *Ha: Hypericum aegyptum*; *Pa: Periploca angustifolia*; *Es: Echinops spinosus*; *Pm: Prasium majus*; *Ap: Atriplex halimus*; H_2_O: H_2_O phase; HE: *n*-hexane extract; MC: CH_2_Cl_2_ extract.

**Figure 3 plants-11-03447-f003:**
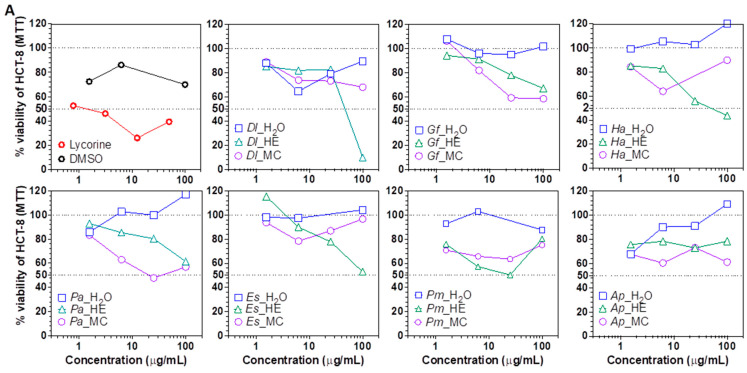
Cytotoxicity of plant extracts. (**A**) Viability of colorectal adenocarcinoma HCT-8 cells treated with plant extract fractions. (**B**) Viability of hepatocarcinoma Huh7 cells treated with plant extract fractions. Two dashed lines were added at CC_50_ and CC_0_ (100% of viability) to ease interpretation. *Dl*: *Daucus lopadusanus*; *Gf*: *Glaucium flavum*; *Ha*: *Hypericum aegyptum*; *Pa*: *Periploca angustifolia*; *Es*: *Echinops spinosus*; *Pm*: *Prasium majus*; *Ap*: *Atriplex halimus*; H_2_O: H_2_O phase; HE: *n*-hexane extract MC: CH_2_Cl_2_ extract; MTT: tetrazolium salt.

**Figure 4 plants-11-03447-f004:**
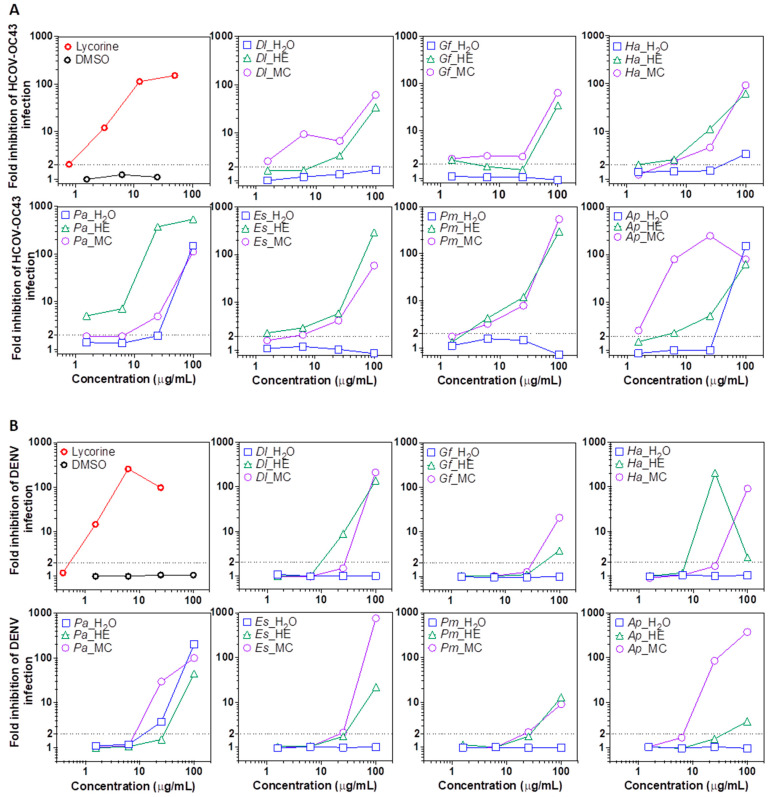
Antiviral activity of plant extract fractions. (**A**) Anti-betacoronavirus 1 (HCOV-OC43, human coronavirus OC43). (**B**) Anti-flavivirus (dengue virus, DENV). Results are expressed as fold inhibition calculated as the ratio of infection (% GFP+ cells) for solvent-treated cells over extract-treated cells. The % of infected (GFP+ cells) was measured by flow cytometry. *Dl*: *Daucus lopadusanus*; *Gf*: *Glaucium flavum*; *Ha*: *Hypericum aegyptum*; *Pa*: *Periploca angustifolia*; *Es*: *Echinops spinosus*; *Pm*: *Prasium majus*; *Ap*: *Atriplex halimus*; H_2_O: H_2_O phase; HE: hexane extract; MC:_:_ CH_2_Cl_2_ extract; MTT: tetrazolium salt.

**Figure 5 plants-11-03447-f005:**
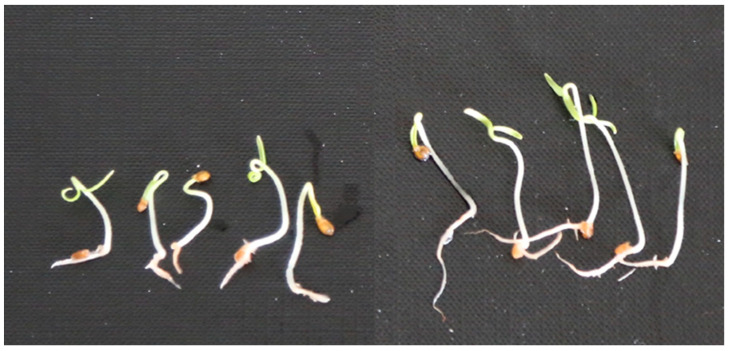
Inhibition of rootlet elongation of tomato plants induced by *Es* extract (**left**). Control (distilled H_2_O, **right**).

**Figure 6 plants-11-03447-f006:**
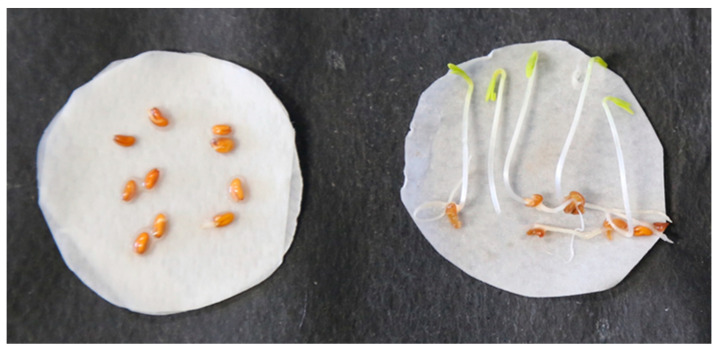
Inhibition of rootlet elongation of cress plants induced by *Es* extract (**left**). Control (distilled H_2_O, **right**).

**Table 1 plants-11-03447-t001:** Name, yields of organic extracts and relevant bibliography for seven native plants collected in Lampedusa.

Plant	Botanical Family	Weight (mg) of ^1^ *HE* Extract	Weight (mg) of MC Extract	N articles Found	N Articles Found with “Metabolite” as Keyword
*Atriplex halimus* (*Ap*)	Amaranthaceae	14.74	40.68	401	4
*Daucus lopadusanus* (*Dl*)	Apiaceae	153.20	84.50	1	0
*Echinops spinosus* (*Es*)	Asteraceae	26.72	119.63	40	0
*Glaucium flavum* (*Gf*)	Papaveraceae	203.57	2765.62	221	0
*Hypericum aegypticum* (*Ha*)	Hypericaceae	77.92	173.96	9	0
*Periploca angustifolia* (*Pa*)	Apocynaceae	12.96	89.28	22	0
*Prasium majus* (*Pm*)	Lamiaceae	71.25	264.23	28	3

^1^ The weight is referred to 100 g of total fresh plant extracted.

**Table 2 plants-11-03447-t002:** Effect of the extracts in phytotoxicity assay.

	Tomato			Cress						Broomrape		
	Rootlet Length (cm)			Germinated Seeds (n.)			Rootlet Length (cm)			Germinated Seeds (%)		
	WR ^1^	MC	HE	WR	MC	HE	WR	MC	HE	WR	MC	HE
*Daucus lopadusanus*	1.8	2.0	3.1	7	4.5	8.5	1.30	1.06	1.42	78.5	56	46.5
*Glaucium flavum*	2.4	2.4	2.8	8	6.5	8	1.10	1.21	1.10	67.5	37	68
*Hypericum aegypticum*	2.9	1.6	3.2	7.5	6.5	5	2.03	1.25	1.13	80	44.5	59
*Periploca angustifolia*	2.5	1.3	2.0	8	5	6	0.88	1.03	0.94	65	44	59.5
*Echinops ritro*	2.4	0.6	2.8	5.5	1	9.5	0.58	0.35	1.07	63.5	34	53
*Prasium majus*	2.4	0.9	3.0	7.5	7.5	5.5	1.55	0.82	1.15	73	47	66
*Atriplex halimus*	1.6	1.1	2.5	7	3	6.5	0.79	0.74	1.13	52.5	54	64.5
Control		3.0			7.5			2.56			70	

^1^ WR, Water residue; MC, Methylene chloride extract; HE, *n*-hexane extract.

**Table 3 plants-11-03447-t003:** Phytotoxic effects of extracts and water residue in the leaf puncture assay.

	CA ^1^			CT			HE			SN			SO			SH			PO		
	WR ^2^	MC	HE	WR	MC	HE	WR	MC	HE	WR	MC	HE	WR	MC	HE	WR	MC	HE	WR	MC	HE
*Daucus lopadusanus*	- ^3^	+	+	-	-	-	-	-	-	-	-	-	-	-	+	-	-	+	-	+	+
*Glaucium flavum*	-	-	+	-	-	-	-	+	-	-	-	-	-	+	-	-	+	+	-	+	+
*Hypericum aegypticum*	-	+	-	-	-	-	-	-	-	-	-	-	-	+	+	-	+	+	-	+	+
*Periploca angustifolia*	-	+	-	-	-	-	+	-	-	-	-	-	-	-	-	-	-	-	-	+	+
*Echinops ritro*	-	+	-	-	-	-	-	-	-	-	-	-	-	-	-	-	+	-	-	+	+
*Prasium majus*	-	-	-	-	-	-	-	-	-	-	-	-	-	-	+	-	-	+	-	-	+
*Atriplex halimus*	-	-	+	-	-	-	-	+	-	-	-	-	-	-	+	-	-	-	-	-	+

^1^ CA-Chenopodium album CT-Chrozophora tinctoria HE-Heliotropium europaeum SN-Solanum nigrum. SO-Sonchus oleraceus SH-Sorghum halepense PO-Portulaca oleracea. ^2^ WR—Water residue; MC-methylene chloride extract; HE-n-hexane extract. ^3^ - = inactive; + = active, production of necrosis.

## Data Availability

Not applicable.

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
