# Peer review of "Biochemical Analyses of Bioactive Extracts from Plants Native to Lampedusa, Sicily Minor Island"

_plants, 2022, doi:10.3390/plants11243447_

Round 1

Reviewer 1 Report

1. Please providing the accurate time for sampling instead of rough description as the preflowering stage, and adding the 7 sample pictures in supplementary. Additionally, only 3 vague material pictures lacking description and illustration were presented in the supplementary, while 7 materials were used in this manuscript.

2. In Materials and Methods, the author did not provide information in detail, including the detection wavelength of HPLC and concrete test parameters of MS. It is vital for detecting the compounds. And also there are no the chromatographic profiles, experimental data and analysis of 7 plants in supplementary material, which cannot support “the results showed the presence of several metabolites in both organic extracts and there are noteworthy differences in the nature of the metabolites produced by the different plants.”. Additionally, it is recommended that you should select control reasonably. For example, the solvent of extractive was methanol in phytotoxicity assay, while control was water, which is not enough to support the result “the extracts from Er and Gf were able to reduce germination of the stimulated seeds by approximately 50% compared with the control”. Finally, the replicates should not less than 3.

3. Please using the consistent and standard units, such as percentage instead of %, and uM or ug/ml in line 308?

4. In fig 1, please adding the negatice control, otherwise, it cannot explained that the extract has inhibition activity. And there was no data in detail, standard errors and difference significance analysis.

5. Selecting the proper threshold in fig 2.

6. Since the antiviral activity of the positive control increasing firstly and then decreasing in fig 3, it is recommend that you could increase the concentration to observe the concentration of the strongest resistance.

7. Methylene chloride extract was described as MC, DC, DMC in table 2? Bold font or not in table 3? Selecting the appropriate size table. There are two conclusions in the manuscript.

8. Checking the grammer errors, such as line 55, line 263 and so on.

Reviewer 2 Report

Dear author,

Kindly give more details/explanations about the following questions:

1- The title of the paper doesn’t fit exactly with the content of article. In fact, in title you mentioned: bioactive metabolites, but I think is better to change it by bioactive extracts because there is no single metabolite was identified despite you did HPLC-MS and you give the chromatograms in supplementary data.

2- The introduction does not provide sufficient background and do not include all relevant references, don’t provide good enough useful information for the readers and should be more updated and more developed. kindly develop more this section. You conducted several experiments related to agriculture and medicine fields, but there are only 13 lines as introduction. I think is too short and is not enough.

3- Kindly revise the information based on literature data in the table 1, ex:  you said that: He, seemed also interesting, as only one published study was reported but any bioactive metabolite was characterized. But in the table 1 you mentioned 3 papers? Also, you mentioned that for tow others extracts there is no paper describe metabolites in these two plants, but I am not 100% sure about that

4- You conducted HPLC-MS, you explain well this method in material and methods section, and you give all details of the technic. You provide the chromatograms in supplementary data, but no compounds were identified in this paper (5 line only: Line 118-122)? Kindly insert in results and discussion section examples of chromatograms and give more information regarding the identified compounds. 

5- The discussion is not well developed, you do not discus all the results that you obtained with previous works, in many paragraphs you described the results and there is no single reference. In total in results and discussion section there is 18 references, distributed as follow:

From Line 72-86: No references

From line 95-104: 7 references (10 lines)

From Line 87-89: 4 references (3 lines)

From Line 108-149 No references

From Line 157-203: No references

Line 204: 4 references (one line)

From Line 205-224: No references, No discussion with previous works (Bioassays on seeds, Leaf puncture assay, Antifungal bioassays)? 

6- You perform many biological essays related to agriculture, kindly provide us some relevant photos for those experiments.

7- Table 3 is not well illustrated, kindly restructure it.

In the following, there is a list of corrections that you should consider:

1-    Line 58: added space after [24,25]

2-    Line 72 to 74: scientific names should be Italic

3-    Line 154: betacoronavirus in the place of betaoronavirus

4-    Line 256: remove line

5-    Line 376: remove space after: (water).

Round 2

Reviewer 2 Report

Dear Author

Thank you very much for your answers.